# Whole-exome sequencing study identifies rare variants and genes associated with intraocular pressure and glaucoma

Xiaoyi Raymond Gao [1,2,3,4] ✉, Marion Chiariglione[1] & Alexander J. Arch[1]

Elevated intraocular pressure (IOP) is a major risk factor for glaucoma, the leading cause of irreversible blindness worldwide. IOP is also the only modifiable risk factor for glaucoma. Previous genome-wide association studies have established the contribution of common genetic variants to IOP. The role of rare variants for IOP was unknown. Using whole exome sequencing data from 110,260 participants in the UK Biobank (UKB), we conducted the largest exome-wide association study of IOP to date. In addition to confirming known IOP genes, we identified 40 novel rare-variant genes for IOP, such as *BOD1L1*, *ACAD10* and *HLA-B*, demonstrating the power of including and aggregating rare variants in gene discovery. About half of these IOP genes are also associated with glaucoma phenotypes in UKB and the FinnGen cohort. Six of these genes, i.e. *ADRB1*, *PTPRB*, *RPL26*, *RPL10A*, *EGLN2*, and *MTOR*, are drug targets that are either established for clinical treatment or in clinical trials. Furthermore, we constructed a rare-variant polygenic risk score and showed its significant association with glaucoma in independent participants (n = 312,825). We demonstrated the value of rare variants to enhance our understanding of the biological mechanisms regulating IOP and uncovered potential therapeutic targets for glaucoma.

Elevated intraocular pressure (IOP) is a major risk factor for glaucoma, the leading cause of irreversible blindness worldwide. IOP is also the only modifiable risk factor for glaucoma. Current glaucoma drugs target lowering IOP. Previous studies using directly genotyped and imputed genetic data have uncovered common and some low-frequency variants for IOP[1–6]. Identifying rare variants that contribute to IOP will help uncover the biological mechanisms regulating this trait and provide improved understanding of IOP regulation and potential therapeutic targets for managing IOP and glaucoma.

Genome-wide association studies (GWAS) have identified over 190 genetic loci associated with IOP[1–4]. These loci have established the contribution of common variants to IOP. The bivariate genetic correlation between IOP and glaucoma was also found to be high (0.49)[7]. While these studies identified numerous loci associated with IOP, these common variants typically show small effect sizes. The role of rare variants for IOP remains to be discovered. Rare variants typically require sequencing and a large sample size to have adequate statistical power.

The UK Biobank (UKB) is a large prospective cohort of half a million adult participants with extensive genetic data linked to physical measurements, health records, family history, and lifestyle information[8]. The recent release of whole-exome sequencing (WES) data now enables the exploration of rare variants for a variety of human traits and diseases and drug targets[9,10], including IOP and glaucoma. Rare variants can have large effect sizes and have demonstrated greater translational potential, e.g., *PCSK9* as a target for lowering low-density lipoprotein levels[11,12] and *MYOC* as a target for gene therapy for treating myocilin-associated glaucoma[13]. These

[1]Department of Ophthalmology and Visual Sciences, The Ohio State University, Columbus, OH 43210, USA. [2]Department of Biomedical Informatics, The Ohio State University, Columbus, OH 43210, USA. [3]Division of Human Genetics, The Ohio State University, Columbus, OH 43210, USA. [4]Ohio State University Physicians Inc., Columbus, OH, USA. ✉e-mail: raymond.gao@osumc.edu

WES variants are also easier to interpret because they directly map to genes.

Using WES data from 110,260 participants in UKB, we conducted an exome-wide association study (ExWAS) to identify rare variants and genes associated with IOP, evaluated their effects on glaucoma in UKB and the FinnGen cohort, and explored potential drug targets of the identified genes. We also constructed a rare-variant polygenic risk score (rvPRS) and tested its association with glaucoma in independent white participants ($n = 312,825$). To the best of our knowledge, this study represents the largest rare-variant study of IOP to date. Our results uncovered rare variants regulating IOP, and subsequently, furthered our understanding of the biological mechanisms of IOP and potential drug targets for managing glaucoma.

## Results

A total of 110,260 UKB participants were included in the IOP WES analysis, of which 98,674 were white. The mean (standard deviation [SD]) of age was 58 (8.1) years and 54% of the participants were female. The average IOP (SD) was 16.0 (3.4; range: 7.0–39.0) mmHg.

We identified 13 rare variants (10 of which are previously unreported) significantly associated with IOP, among which six were identified in white-only (white participants extracted based on a combination of self-reported White ethnicity [UKB data field 21000] and genetic information, see Methods for details) analysis and seven additional ones were identified in pan-ancestry (all ancestry combined) analysis. Table 1 displays the single-variant association results. Our top SNP, rs74315329 ($P = 1.22 \times 10^{-26}$) is a well-known stop-gain variant in *MYOC*, the first gene identified for primary open-angle glaucoma (POAG)[14]. Consistently, rs28991009 in *ANGPTL7* previously identified in our array-based GWAS[2] shows significance in this ExWAS using WES data. In white-only results, rs37278669, a nonsynonymous variant (allele frequency [AF] = 0.011%), in *BOD1L1*, shows a significant association with IOP ($P = 5.75 \times 10^{-9}$, beta = 4.08) in UKB. *BOD1L1* is also significantly associated with the FinnGen phenotype "use of anti-glaucoma preparations and miotics" ($P = 7.7 \times 10^{-6}$). The start-loss variant, rs753877638, in *ACAD10* is significantly associated with both IOP ($P = 1.30 \times 10^{-10}$, beta = 8.41, AF = 0.003%) and glaucoma ($P = 3.68 \times 10^{-4}$) in UKB. In pan-ancestry analysis, rs201956837 in *HLA-B* is associated with IOP ($P = 8.65 \times 10^{-9}$, beta = 4.37). Rs201956837 is an intronic variant as well as an upstream transcript variant. The gene *HLA-B* is highly associated with glaucoma in FinnGen ($P = 8.0 \times 10^{-9}$). *BOD1L1*, *RALYL*, *LDB3*, *ACAD10*, *CDK11A*, and *DPF3* are also associated with glaucoma topical treatments ($P < 1 \times 10^{-5}$, details in Supplementary Data 1). A Manhattan plot of the genome-wide $P$ values for pan-ancestry results is shown in Fig. 1a. The genomic control lambda for white-only and pan-ancestry analyses are 1.01 and 1.02, respectively, which are well under control. The corresponding quantile-quantile plots are shown in Supplementary Fig. 1.

From SAIGE-GENE analysis, 35 additional genes showed significant associations with IOP and 31 of them not previously published, among which 11 were identified from white-only analysis and 20 additional ones were identified from pan-ancestry analysis. Table 2 displays the gene-based association results. A Manhattan plot of the genome-wide p-values for pan-ancestry results is shown in Fig. 1b. Rare variants in previously known IOP genes, *MTOR*[2], *EVA1C*[15], and *CFAP298-TCP10L*[15], identified from common-variant investigations show significant gene-based associations with $P = 1.08 \times 10^{-12}$, $P = 9.51 \times 10^{-10}$, and $P = 1.34 \times 10^{-8}$, respectively. Several of these ExWAS significant IOP genes, such as *PTPRB*, *KIF21A*, *DNTT*, also show a significant association with glaucoma in UKB with $P = 3.26 \times 10^{-5}$, 0.009, 0.007, respectively. Many of these IOP genes are associated with glaucoma-related traits in FinnGen. For example, *CDCA8*, *HLA-B*, *RHOC*, *PPM1J*, *RPL10A*, and *TEAD3* are associated with glaucoma ($P < 1 \times 10^{-6}$). *ADRB1*, *AAK1*, *IFI27*, *SYNGR3*, and *ZNF598* are associated with POAG ($P < 1 \times 10^{-5}$). Twelve of these genes, including *PTPRB*, *HFM1*, *TAF1B*, *AAK1*, *FOXD1*, *EHMT1*, and

*DNTT*, are associated with glaucoma topical treatments ($P < 1 \times 10^{-5}$, details in Supplementary Data 1).

To seek biological support for the identified genes, we evaluated their gene expression using both bulk RNA and single-cell RNA (scRNA) expression datasets. Supplementary Fig. 2 displays the bulk RNA expression information from Genevestigator[16]. A number of genes, such as *BOD1L1*, *HLA-B*, *RPL10A*, and *RAB4B-EGLN2*, are highly expressed in the trabecular meshwork (TM). Several other genes, e.g., *ACAD10* and *DNTT*, show a medium gene expression in TM. Supplementary Figs. 3 and 4 display the scRNA expression information from the Cell atlas of the human ocular anterior segment (OAS)[17] and of aqueous humor outflow pathways (AHOP)[18], respectively. IOP and glaucoma related cell types can include TM fibroblasts, Schlemm canal endothelium (SCE), ciliary muscle (CM), corneal endothelium (CE), and vascular endothelium (VE)[17,18]. Most of the identified genes show various levels of expression in these cell types. For example, *BOD1L1* is expressed in all the above cell types; *HLA-B* and *PTPRB* are expressed in SCE and VE; and *ACAD10* is expressed in CM and CE; and *RALYL* is expressed in CM, CE, and TM fibroblasts, to name a few.

To query potential drug targets, we used the Open Targets online resource. Table 3 displays the current known and proposed drug targets for these IOP rare-variant genes, such as *ADRB1* (adrenoceptor beta 1, identified from our gene-based analysis), which is a known drug target for topical beta-adrenergic receptor antagonists, or beta-blockers, known to lower IOP. To the best of our knowledge, our results provided evidence for an unreported association between IOP and *ADRB1*. *ADRB1* is expressed in human TM and ciliary body[19], as well as cardiac tissue (Supplementary Fig. 2). Glaucoma drugs targeting *ADRB1* include topical beta-blockers, such as timolol, betaxolol, carteolol, levobunolol, levobetaxolol, and metipranolol. Two older, outdated glaucoma medications include the adrenergic agonists, dipivefrin and epinephrine. In addition, several of these drugs are also used in treating hypertension and cardiovascular disease. *PTPRB* is highly expressed in the vein and artery endothelium cells (Supplementary Fig. 2). It is a proposed drug target for retinal vein occlusion, diabetic retinopathy and diabetic macular edema. Razuprotafib is a small molecule targeting *PTPRB* that acts as a negative regulator of Tie2 in diseased vascular endothelium by receptor-type tyrosine-protein phosphatase beta inhibition. *EGLN2*, a neighboring gene from the readthrough gene *RAB4B-EGLN2*, has drug trials for roxadustat, daprodustat, and vadadustat, which inhibit a hypoxia-inducible factor prolyl hydroxylase. These drugs target anemia and chronic kidney disease. *MTOR* is targeted by perhexiline, a drug used for cardiovascular disease that inhibits the serine/threonine-protein kinase mTOR. The *MTOR* gene is highly expressed in microvessel endothelium cells throughout the eye (Supplementary Fig. 2). *RPL26* and *RPL10A* have three experimental drugs, i.e., ataluren, ELX-02, and MT-3724, two of which (ataluren and ELX-02) work as 80S ribosome modulators while MT-3724 functions as an 80S inhibitor. These drugs are in development for various diseases, such as cystic fibrosis, muscular dystrophy, hemophilia, epilepsy, kidney disease, and leukemia. Drug target genes *ADRB1*, *PTPRB*, and *RPL10A*, among others, were additionally found to have associations with vascular related phenotypes through PheWeb (Supplementary Data 2).

We further constructed a rare-variant polygenic risk score (rvPRS) using the IOP rare variants with $P < 5 \times 10^{-7}$ from pan-ancestry analysis (Supplementary Table 1) and tested its association with glaucoma in independent UKB white individuals ($n = 312,825$), who did not participate in the IOP measurements. This rvPRS is significantly associated with glaucoma with odds ratio (OR) per SD = 1.12 and $P = 5.13 \times 10^{-16}$, indicating the relevance of these IOP rare variants in glaucoma. When we used the rare variants identified from white-only subjects, the rvPRS yielded a mitigated association with glaucoma with OR per SD = 1.07 and $P = 2.18 \times 10^{-8}$. Since the IOP heritability explained by WES rare variants is less than 2% (estimated using GCTA[20,21]), the overall

**Table 1 | Exome-wide significant rare variants for intraocular pressure**

| | Chr | Pos | rsID | AO | A1 | A1 freq | Beta | P | Gene | Function | UKB Glaucoma | PhenoScanner | FinnGen Glaucoma |
|---|---|---|---|---|---|---|---|---|---|---|---|---|---|
| | | | | | | | | | | | $P$ | $P_{GTT}/P_S$† | $P_G/P_P$†$/P_M$†† |
| White | 1 | 11193627 | rs28991009 (p.Gln175His) | G | T | 0.796% | -0.53 | 1.19E-10 | ANGPTL7 | Nonsynonymous | 2.09E-06 | – | 1.0E-24 |
| | 1 | 171636338 | rs74315329 (p.Gln368Ter) | G | A | 0.134% | 2.12 | 1.22E-26 | MYOC | Stop gain | 1.01E-37 | 4.98E-15† | 1.2E-25† |
| | 4 | 13613559 | rs372786669 (p.Glu426Gly) | T | C | 0.011% | 4.08 | 5.75E-09 | **BOD1L1** | Nonsynonymous | 0.073 | 4.26E-7 | 7.7E-6†† |
| | 8 | 84887679 | rs371413262 (p.Ser267Phe) | C | T | 0.003% | 7.73 | 3.66E-09 | **RALYL** | Nonsynonymous | – | 1.43E-11 | 8.1E-5†† |
| | 10 | 86687010 | rs112082622 | C | T | 0.003% | 8.62 | 1.83E-09 | **LDB3** | Intronic | – | 2.62E-7 | 1.5E-4† |
| | 12 | 111692710 | rs753877638 (p.Met1Val) | A | G | 0.003% | 8.41 | 1.30E-10 | **ACAD10** | Start loss | 3.68E-04 | 7.16E-6 | 1.0E-3 |
| Pan-ancestry (additional hits) | 1 | 1708342 | rs556417493 | C | G | 0.003% | 6.55 | 8.67E-09 | **CDK11A** | Intronic | 0.023 | 5.66E-6 | 5.7E-5† |
| | 1 | 244432616 | rs375507039 | T | C | 0.003% | 7.69 | 5.00E-09 | **ADSS2** | Intronic, upstream transcript variant | 0.016 | – | 3.6E-5†† |
| | 6 | 31356961 | rs201956837 | G | A | 0.008% | 4.37 | 8.65E-09 | **HLA-B** | Intronic, upstream transcript variant | – | 1.55E-20† | 8.0E-9 |
| | 10 | 73911900 | rs367716060 (p.Glu81*) | C | A | 0.002% | 8.27 | 8.77E-09 | **PLAU** | Nonsynonymous | 0.006 | – | 9.1E-5† |
| | 10 | 79349655 | rs557881342 | C | T | 0.003% | 7.60 | 7.03E-09 | **PPIF** | Intronic | – | – | 3.1E-7 |
| | 14 | 72670683 | rs933632776 | AC | A | 0.002% | 8.99 | 3.92E-10 | **DPF3** | UTR3 | – | 3.79E-14 | 8.1E-5 |
| | 19 | 1529427 | rs776910868 (p.Val143Phe) | G | T | 0.004% | 6.93 | 1.09E-09 | **PLK5** | Nonsynonymous | – | – | 8.4E-5† |

REGENIE was used to perform single-variant association tests (two-sided). Rare variants for intraocular pressure with $P < 1 \times 10^{-8}$ are presented. Their corresponding association results for glaucoma-related traits in UK Biobank, PhenoScanner and FinnGen are also shown. To facilitate visualization, $P$ values that do not pass a pre-defined cutoff, 0.1 for UKB Glaucoma and $1 \times 10^{-5}$ for PhenoScanner, are displayed as a dash (–). The upper panel shows white-only significant results. The bottom panel shows additional hits from pan-ancestry analysis. No adjustments were made for multiple comparisons. Gene name is in boldface if it has not been previously reported for intraocular pressure. Genomic positions are according to hg38.

Chr chromosome, Pos position, AO allele 0, A1 allele 1, A1Freq allele 1 frequency in the analyzed sample, UKB UK Biobank.

$P_{GTT}$, $P$ value for glaucoma topical treatment from PhenoScanner; $P_S$,† $P$ value for other serious eye conditions from PhenoScanner; $P_G$, $P$ value for glaucoma in FinnGen; $P_P$,† $P$ value for primary open-angle glaucoma in FinnGen; $P_M$,†† $P$ value for antiglaucoma preparations and miotics in FinnGen.

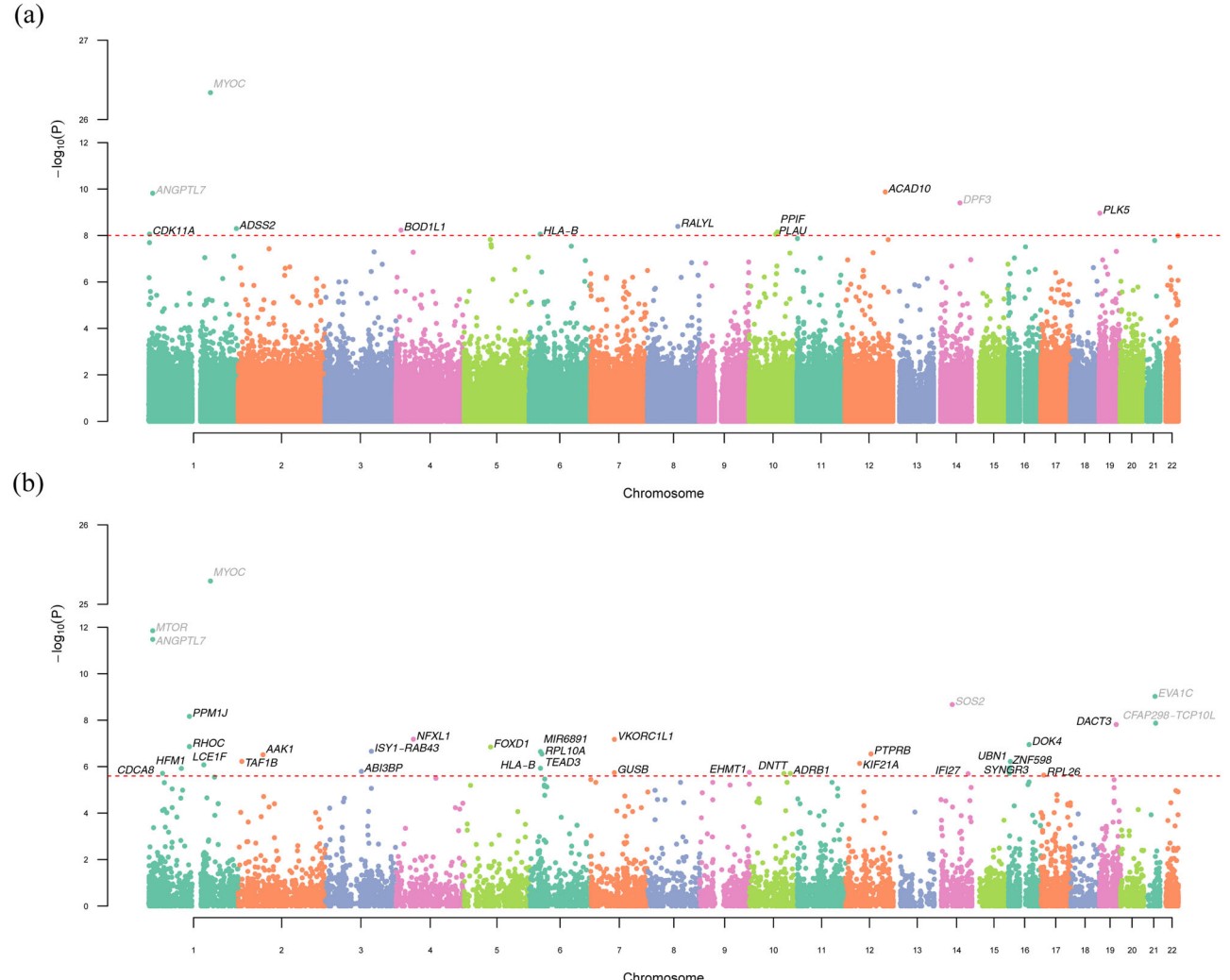

**Fig. 1 | Manhattan plots displaying the −log₁₀(P) for the association between IOP and rare variants and genes.** **a** Single-variant pan-ancestry results. The dotted horizontal line represents exome-wide significance associations ($P < 1 \times 10^{-8}$). **b** Gene-based pan-ancestry results. The dotted horizontal line represents gene-based significance associations ($P < 2.5 \times 10^{-6}$). Genetic variants or genes are plotted by genomic position. The colors on both plots show the delimitation for chromosome. Gene name is in black text if it has not been previously reported for intraocular pressure. Tests conducted in these analyses were two-sided and no adjustments were made for multiple comparisons.

prediction improvement over the baseline model (with only age and sex) in terms of the area under the receiver operating characteristic curve (AUC) of the current rvPRS is relatively low, 0.5%, in comparison to more than 5% in AUC improvement from common variants[22], which can explain about 40% of the IOP heritability[2].

## Discussion

In this study, we conducted the largest ExWAS of IOP to date using data from UKB. By employing single-variant and gene-based analyses, two complementary frameworks, we have expanded our knowledge of the genetic architecture of IOP, especially of the role of rare variants, beyond previous studies involving microarray data, which mainly covered common variants. In addition to confirming known IOP genes, we identified 40 previously unreported genes for IOP, demonstrating the power of including and aggregating rare variants in gene discovery. About half of these IOP genes are also associated with glaucoma phenotypes, including glaucoma medications, in UKB and FinnGen. Six of these genes are drug targets that are either established for clinical treatment or in clinical trials. Furthermore, we constructed a rvPRS and showed its significant association with glaucoma in independent white subjects.

We also showed that including subjects of all ancestries in a pan-ancestry analysis further improved the statistical power to identify rare variants. It was evident that pan-ancestry analyses identified additional rare variants and genes beyond white-only analyses in both single-variant and gene-based analyses. Testing these IOP variants and genes for their effects in glaucoma-related traits in both UKB and FinnGen and querying for drug targets further increased their translational relevance. Furthermore, the IOP rvPRS constructed using the rare variants identified from pan-ancestry analysis showed an even stronger association signal with glaucoma in independent white subjects than using white-only rare variants.

A concern with multi-ancestry datasets is false-positive signals. Numerous previous GWAS used European subjects only. In some studies, it was further reduced to unrelated European subjects. One way to analyze multi-ethnic GWAS datasets is using meta-analysis[1,23], which is typically used for dealing with common variants. However, rare variants may not have enough carriers in individual ancestral groups, resulting in too few carriers to be analyzed. A pooled approach is an attractive alternative for combining ancestrally diverse populations[24], especially for rare variants. Recent advances in statistical genetics tools also made this possible. For example,

**Table 2 | Genome-wide significant results for intraocular pressure from gene-based analysis**

| Chr | Pos | Gene | P | UKB Glaucoma P | PhenoScanner $P_{GTT}/P_{AT}$[†] | FinnGen Glaucoma $P_G/P_P$[†]$/P_M$[††] |
|---|---|---|---|---|---|---|
| **White** | | | | | | |
| 1 | 11106534–11273497 | *MTOR* | 1.08E-12 | 1.03E-05 | – | 1.0E-24 |
| 1 | 11189323–11195981 | *ANGPTL7* | 4.59E-13 | 1.95E-06 | – | 1.0E-24 |
| 1 | 37692515–37709719 | **CDCA8** | 1.13E-06 | – | – | 7.0E-7 |
| 1 | 171635416–171652688 | *MYOC* | 2.06E-25 | 3.22E-36 | 4.98E-15 | 1.2E-25[†] |
| 1 | 183471992–183554193 | **SMG7** | 1.37E-06 | – | 2.09E-6 | 1.5E-4[†] |
| 4 | 47847232–47914667 | **NFXL1** | 7.34E-07 | 0.028 | 3.34E-6[†] | 4.9E-4[††] |
| 6 | 31353874–31357179 | **HLA-B** | 1.24E-06 | – | – | 8.0E-9 |
| 6 | 31355223–31355316 | **MIR6891** | 1.33E-07 | – | – | – |
| 10 | 114043865–114046904 | **ADRB1** | 1.16E-06 | 0.085 | – | 5.8E-6[†] |
| 12 | 39293227–39443147 | **KIF21A** | 7.31E-07 | 0.009 | – | 6.7E-4[†] |
| 12 | 70515869–70637440 | **PTPRB** | 2.80E-07 | 3.26E-05 | 1.50E-10 | 5.9E-5[†] |
| 16 | 57471921–57487139 | **DOK4** | 4.01E-07 | – | – | 5.1E-4 |
| 17 | 8377515–8383193 | **RPL26** | 2.18E-06 | – | – | 1.1E-4[†] |
| 19 | 40778218–40808441 | **RAB4B-EGLN2** | 1.93E-06 | – | – | 4.8E-5[†] |
| **Pan-ancestry (additional hits)** | | | | | | |
| 1 | 91260765–91408008 | **HFM1** | 1.20E-06 | – | 2.05E-8 | 9.1E-4[†] |
| 1 | 112701126–112707403 | **RHOC** | 1.36E-07 | – | – | 2.9E-7 |
| 1 | 112709993–112715328 | **PPM1J** | 6.82E-09 | – | – | 4.3E-7 |
| 1 | 152776371–152776969 | **LCE1F** | 8.35E-07 | – | – | 1.6E-5 |
| 2 | 9843441–9934416 | **TAF1B** | 5.95E-07 | – | 1.56E-9 | 5.3E-5[†] |
| 2 | 69457996–69643739 | **AAK1** | 3.06E-07 | 0.085 | 7.44E-8 | 8.8E-6[†] |
| 3 | 100749328–100993508 | **ABI3BP** | 1.61E-06 | – | – | 3.3E-4 |
| 3 | 129087568–129161230 | **ISY1-RAB43** | 2.19E-07 | – | 3.34E-6[†] | 2.5E-3[†] |
| 5 | 73446265–73448777 | **FOXD1** | 1.43E-07 | – | 7.11E-7 | 6.5E-5 |
| 6 | 35468400–35470781 | **RPL10A** | 2.89E-07 | 0.031 | | 6.0E-7 |
| 6 | 35473596–35497079 | **TEAD3** | 2.84E-07 | 0.033 | 4.63E-6[†] | 6.0E-7 |
| 7 | 65865771–65959558 | **VKORC1L1** | 6.72E-08 | 0.023 | 7.46E-7 | 3.3E-4[†] |
| 7 | 65960683–65982230 | **GUSB** | 1.82E-06 | 0.061 | 3.66E-6[†] | 3.3E-4[†] |
| 9 | 137619000–137836127 | **EHMT1** | 1.77E-06 | – | 2.92E-8 | 1.7E-4[†] |
| 10 | 96304433–96338564 | **DNTT** | 1.99E-06 | 0.007 | 1.23E-7 | 1.1E-5[†] |
| 14 | 50117129–50231578 | *SOS2* | 2.13E-09 | – | 6.44E-9 | 6.1E-4 |
| 14 | 94110735–94116690 | **IFI27** | 2.02E-06 | – | – | 9.2E-6[†] |
| 16 | 1989969–1994275 | **SYNGR3** | 1.93E-06 | – | – | 3.3E-7[†] |
| 16 | 1997653–2009821 | **ZNF598** | 1.14E-06 | – | – | 3.3E-7[†] |
| 16 | 4846664–4882401 | **UBN1** | 6.05E-07 | – | 1.13E-6 | 5.2E-5[†] |
| 19 | 46647550–46661182 | **DACT3** | 1.54E-08 | – | 2.65E-6[†] | 8.9E-5[†] |
| 21 | 32411691–32515387 | **EVA1C** | 9.51E-10 | – | 3.85E-6 | 7.8E-5[†] |
| 21 | 33935802–33984687 | *CFAP298-TCP10L* | 1.34E-08 | – | – | 7.8E-5[†] |

SAIGE was used to perform gene-based association tests (two-sided). Genes with $P < 2.5 \times 10^{-6}$ for intraocular pressure are presented. Their corresponding association results for glaucoma-related traits in UK Biobank, PhenoScanner, and FinnGen are also shown. To facilitate visualization, P values that do not pass a pre-defined cutoff, 0.1 for UKB Glaucoma and $1 \times 10^{-5}$ for PhenoScanner, are displayed as a dash (–). The upper panel shows white-only significant results. The bottom panel shows additional hits from pan-ancestry analysis. No adjustments were made for multiple comparisons. Gene name is in boldface if it has not been previously reported for intraocular pressure. Genomic positions are according to hg38.
*Chr* chromosome, *Pos* position, *UKB* UK Biobank.
$P_{AT}$[†], P value for artificial tear medication from PhenoScanner; $P_G$, P value for glaucoma in FinnGen; $P_P$[†], P value for primary open-angle glaucoma in FinnGen; $P_M$[††], P value for antiglaucoma preparations and miotics in FinnGen.

REGENIE[25], a machine learning approach, can avoid the parameter inflation in ultra-rare-variant situations while controlling for both population stratification and sample relatedness. SAIGE[26] uses mixed-effects models to adjust for both population stratification and genetic relationship matrix. Using these state-of-the-art methods, we showed the effectiveness of including non-European subjects in pan-ancestry analyses to further increase the study power. With advanced statistical genetics tools that can adjust for both genetic relatedness, principal components (PCs) of genetic ancestry and ancestral clusters, it is feasible to carry out pan-ancestry analyses in a pooled/

combined approach. To further validate the robustness of our pan-ancestry results, we analyzed the significant rare variants in several sub-populations (UKB data field 21000), i.e. Black, Asian, and non-white all together, if there were sufficient allele counts for analysis, i.e., minor allele count (MAC) ≥ 5 (REGENIE default), in each sub-population. Supplementary Table 3 and Supplementary Data 3 show the sub-population-specific allele counts and association results, respectively. *CDK11A* rs556417493 is associated with IOP in Asian participants (AF = 0.08%, beta = 6.51, $P = 7.59 \times 10^{-9}$). *PLK5* rs776910868 is associated with IOP in Black participants (AF = 0.1%,

**Table 3 | Known drug targets for the identified intraocular pressure rare-variant genes**

| Gene | Known drugs | Mechanism of action | Disease information (including those in clinical trials) |
|---|---|---|---|
| *ADRB1* | CARTEOLOL HYDROCHLORIDE[a] | Beta-adrenergic receptor antagonist | Open-angle glaucoma |
| | TIMOLOL[a] | Beta-adrenergic receptor antagonist | Ocular hypertension, low tension glaucoma, cardiovascular disease, glaucoma, hemangioma, open-angle glaucoma, eye disease, corneal edema, portal hypertension, exfoliation syndrome, migraine disorder, varicose disorder, wet macular degeneration, hereditary hemorrhagic telangiectasia, diabetic macular edema, anterior ischemic optic neuropathy |
| | BETAXOLOL HYDROCHLORIDE[a] | Beta-1 adrenergic receptor antagonist | Open-angle glaucoma, hypertension, ocular hypertension |
| | TIMOLOL MALEATE[a] | Beta-1 adrenergic receptor antagonist | Glaucoma, ocular hypertension, hemangioma, open-angle glaucoma, corneal edema, portal hypertension, ocular hypertension, anterior ischemic optic neuropathy |
| | CARTEOLOL | Adrenergic receptor beta antagonist | Glaucoma, cardiovascular disease, open-angle glaucoma |
| | DIPIVEFRIN[a] | Adrenergic receptor agonist | Glaucoma (no longer available) |
| | LEVOBETAXOLOL HYDROCHLORIDE[a] | Beta-1 adrenergic receptor antagonist | Glaucoma, ocular hypertension |
| | EPINEPHRINE[a] | Adrenergic receptor agonist | Glaucoma (no longer available) |
| | BETAXOLOL | Beta-1 adrenergic receptor antagonist | Hypertension, cardiovascular disease, open-angle glaucoma, ocular hypertension |
| | METIPRANOLOL | Beta-1 adrenergic receptor antagonist | Glaucoma |
| | LEVOBUNOLOL | Beta-1 adrenergic receptor antagonist | Glaucoma |
| | LEVOBETAXOLOL | Beta-1 adrenergic receptor antagonist | Glaucoma, ocular hypertension |
| *PTPRB* | RAZUPROTAFIB | Receptor-type tyrosine-protein phosphatase beta inhibitor | Diabetic macular edema, non-proliferative diabetic retinopathy, retinal vein occlusion, glaucoma (Brigell et al.[31]) |
| *RPL26* *RPL10A* | ATALUREN | 80S Ribosome modulator | Cystic fibrosis, Duchenne muscular dystrophy, Becker muscular dystrophy, aniridia, disorder of amino acid and other organic acid metabolism, hemophilia B, hemophilia A, epilepsy |
| | ELX-02 | 80S Ribosome modulator | Genetic disorder, kidney disease, cystinosis, cystic fibrosis |
| | MT-3724 | 80S Ribosome inhibitor | Diffuse large B-cell lymphoma, non-Hodgkin's lymphoma, lymphoid leukemia, chronic lymphocytic leukemia |
| *EGLN2* | ROXADUSTAT | Hypoxia-inducible factor prolyl hydroxylase inhibitor | Chronic kidney disease, anemia, myelodysplastic syndrome, ST elevation myocardial infarction |
| | DAPRODUSTAT | Hypoxia-inducible factor prolyl hydroxylase inhibitor | Anemia, peripheral vascular disease |
| | VADADUSTAT | Hypoxia-inducible factor prolyl hydroxylase inhibitor | Anemia, chronic kidney disease |
| *MTOR*[b] | PERHEXILINE | Serine/threonine-protein kinase mTOR inhibitor | Cardiovascular disease, hypertrophic cardiomyopathy, diabetic cardiomyopathy, diastolic heart failure, heart failure |
| | PALOMID-529 | Serine/threonine-protein kinase mTOR inhibitor | age-related macular degeneration |

This table shows the existing drugs that target the intraocular pressure rare-variant genes identified.
[a]FDA approved treatment for glaucoma and/or ocular hypertension.
[b]*MTOR* has many more drugs and diseases in clinical trials not listed totaling 24 drugs and 87 diseases.

beta = 8.14, $P = 8.42 \times 10^{-10}$). These sub-population-specific results are highly consistent with our pan-ancestry results. *ADSS2* rs375507039, *PLAU* rs367716060, and *DPF3* rs933632776 are associated with IOP in non-white participants and could not be analyzed in separate individual sub-populations due to rare allele counts. For the *HLA-B* variant rs201956837, there is consistent direction of allele effects and effect sizes in white (beta = 4.33, $P = 7.76 \times 10^{-6}$) and non-white (beta = 4.47, $P = 3.02 \times 10^{-4}$). Using Fisher's method to combine the two p-values (white and non-white), we obtained $P = 2.35 \times 10^{-9}$, which is consistent with our pan-ancestry result, $P = 8.65 \times 10^{-9}$, for the variant. The consistency of these results with our pan-ancestry results demonstrates the robustness of our analysis. Further research is warranted to maximize the power of pan-ancestry analysis[24].

Genetics provides vital information to identify drug targets. The generation of this WES data is sponsored by eight pharmaceutical companies, including Regeneron and AstraZeneca[9], which clearly shows the value of this dataset to that industry. Drug candidates that have genetics support are twice as likely to be successful than those without genetics support[27]. Six genes, i.e., *ADRB1*, *PTPRB*, *RPL26*, *RPL10A*, *EGLN2*, and *MTOR*, out of our gene-based analyses have existing therapeutic molecular targets. The most notable one, *ADRB1*, is the target of cardiovascular and glaucoma drugs, which include the broad class of glaucoma drugs targeting the beta-adrenergic receptor antagonists. For example, timolol was a first-line drug for lowering IOP by blocking the beta-adrenergic receptors in the ciliary body[28] to decrease aqueous humor flow[29]. More recently, timolol has been shown to have an effect on outflow facility[30], which also impacts IOP. The other five genes are targets in many clinical trials involving razuprotafib, ataluren, ELX-02, MT-3724, roxadustat, daprodustat, vadadustat, and perhexiline, which provide candidates for drug repurposing for possible glaucoma treatment. For example,

razuprotafib has been shown recently as an adjunct to latanoprost for treating glaucoma patients[31]. Razuprotafib also appears to stabilize blood vessels[31]. Roxadustat has proposed pathways affecting blood cell production[32]. Taken together, many of these drugs appear to be involved with cardiovascular disease and blood flow. Additionally, phenome-wide associations of the identified genes showed numerous significant associations with vascular-related phenotypes (Supplementary Data 2). Validations of cardiovascular relationships and drug targets for these IOP associated genes and recent success with drugs targeting vascular areas for glaucoma as seen with razuprotafib[31] indicate that it may be possible to repurpose certain drugs that work on cardiovascular disease for glaucoma.

This study is not without limitations. Rare variants have their intrinsic challenges. The rarity of these variants makes their replication far more difficult than common variants. Nevertheless, since IOP is an endophenotype of glaucoma and ~70% glaucoma GWAS hits are also associated with IOP[23], it is reasonable to test these IOP hits for their glaucoma effects[4] although it should not be assumed that all IOP hits are associated with glaucoma. Furthermore, IOP hits can also provide translation implications for glaucoma management since lowering IOP is currently the sole proven solution for glaucoma treatment. Hence, we checked the significance of these IOP variants and genes on glaucoma-related traits, including glaucoma treatment medication, in both UKB and FinnGen cohorts. In UKB, a combination of self-reported glaucoma and ICD-10/9 codes for glaucoma phenotypes is not homogeneous for specialized glaucoma subtypes, but previous studies have demonstrated the effect of using it for studying POAG genetics[4,33]. Despite being the largest WES data currently available, diversity is still low, and European subjects comprise about 94% of the UKB cohort. Other larger ancestral groups are no doubt invaluable and can provide further information for discovery and validation. About half of the IOP rare-variant genes identified were found to be associated with glaucoma-related traits in either UKB or FinnGen. Further studies are required to confirm the remaining ones for their impacts on glaucoma. Furthermore, the best approach for analyzing datasets of ancestrally diverse populations remains an ongoing research topic[24], especially for rare variants. We used a combination of self-reported ethnic background and PCs of genetic ancestry to identify white participants who have similar ancestral backgrounds. Despite these efforts, subtle structure may still present. Sub-population labels for our analyses in Black and Asian in Supplementary Tables 3 and Supplementary Data 3 were extracted from self-reported ethnicity (UKB data field 21000). Self-reported ethnic background may be inaccurate for some individuals. However, we used state-of-the-art methods, i.e. REGENIE and SAIGE, which can handle both population structure and cryptic relatedness. There are many different approaches to analyze rare variants, e.g., grouping by predicted loss-of-function (pLOF) variants, missense variants, synonymous variants[34], all inclusive[35], and sliding window[36]. To our knowledge, there is no consensus on the best approach to analyze rare variants. It can be phenotype dependent as well. Using other approaches may identify more rare variants and genes associated with IOP. Our rvPRS showed significant association with glaucoma, demonstrating the aggregated effect of the rare variants on glaucoma. However, its discriminatory ability in glaucoma prediction in terms of AUC is still low, which indicates that rare variants from WES may be more useful for biological insight than prediction at present. An exome only comprises about 1% of the human genome. Whole-genome sequencing data should be able to explain more IOP heritability. Hence, the best strategy to incorporate rare variants in PRS construction warrants further studies.

In conclusion, we carried out the largest ExWAS of IOP to date. In addition to showing the efficacy of single-variant and white-only analyses, our study clearly supports using gene-based aggregation and pan-ancestry analyses to further increase the study power. We demonstrated the value of rare variants to enhance our understanding of the biological mechanisms regulating this trait, and uncovered potential therapeutic targets for glaucoma.

# Methods

## UKB resource
UKB is an ongoing large prospective cohort study. Details regarding this cohort have been described elsewhere[37,38]. Briefly, the UKB recruited over 500,000 adult participants (40–70 years of age at enrollment) living in the United Kingdom who were registered with the National Health Service at the study baseline (from 2006 to 2010). Medical information (self-report and electronic health records), family history, lifestyle information, as well as DNA samples, were collected. Ophthalmological data were also collected for a subset of study participants (~118,000). Most participants (~94%) reported their ethnic background as white and the rest originated outside of Europe[8]. UKB was approved by the North West Multi-Centre Research Ethics Committee and written informed consent was obtained from all participants. Our access to the resource was approved by UKB (application number 23424) and we obtained access to fully de-identified data.

## FinnGen resource
The Finngen study is a large biobank study focused on the population of Finland[39]. Over 200,000 participants have been enrolled, genotyped and phenotyped. 500,000 participants are projected to be enrolled by the end of 2023. The study aims to show the power of nationwide biobanks, electronic health records and an isolated population in identifying rare variants associated with different diseases. Data was collected from different Finnish biobanks and digital health care data on Finland citizens starting in 2017. The recruited population has an age average of 63 years and hospital-based recruitment predominates thus far. Phenotypes were built using the International Classification of Disease Ninth and Tenth Revision (ICD-9 and ICD-10) codes. Genotyping was done with a custom Axiom FinnGen1 and legacy arrays and further imputed to 17 million markers based on whole-genome sequences of Finns. Out of 2861 endpoint phenotypes created for this study, 15 are glaucoma related: neovascular glaucoma, primary angle-closure glaucoma, other and unspecified glaucoma, glaucoma, use of antiglaucoma preparations and miotics, juvenile open-angle glaucoma, normotensive glaucoma, glaucoma-related operations, primary open-angle glaucoma (strict), glaucoma (exfoliation), primary open-angle glaucoma, glaucoma secondary to other eye disorders, glaucoma secondary to eye inflammation, glaucoma secondary to eye trauma, and glaucoma suspect. The study used the SAIGE mixed models for their association analyses. The summary statistics are publicly online available (see data availability).

## UKB WES and quality control
WES for all UKB participants were generated at the Regeneron Genetic Center[9,10]. The sequencing, variant calling, and quality control were detailed previously[9,40]. Briefly, sequencing was done on the Illumina NovaSeq 6000 platform using 75 base pair paired-end reads. Variant calling and quality control were performed using the SPB protocol[41]. The high-quality WES data have been reported to exceed 20× coverage at 95.8% of targeted bases. We overlapped the data with participants who participated in the ophthalmological measurements and kept all samples that had missing rate <2.5%. We kept autosomal variants with call rate >95% and minor allele count (MAC) ≥1 (15.1 million). We annotated these variants using VEP[2] and annovar[42].

## IOP measurements in UKB
IOP measurements were obtained using the Optical Response Analyzer (Reichert Corp., Philadelphia, PA) and have been described previously[43]. In brief, both corneal-compensated and Goldman-correlated IOP measurements were collected. We used corneal-compensated IOP for this study since it is less affected by corneal

thickness[44,45]. The average of both eyes was used for downstream analysis. If only one IOP measurement was obtained, it was used as the final value. Study participants who received eye surgery within 4 weeks prior to the ocular assessment or those with possible eye infections did not receive IOP measurements. Moreover, we excluded study participants with extreme values of IOP, i.e., in the bottom and top 0.3 percentiles, and outliers, including participants who had either eye surgery or used eye drop medications[2,22]. Overlapping with the WES data, 98,674 white participants (based on a combination of self-reported White ethnicity [UKB data field 21000] and genetic information; outliers with genetic ancestry at least six SDs from the means of the first two PCs were removed) and 110,260 pan-ancestry (all ancestry combined) participants remained.

### Single-variant and gene-based ExWAS analyses

We performed single-variant association analyses using a machine-learning method implemented by REGENIE[25], accounting for population stratification and sample relatedness. We analyzed all variants with MAC ≥ 5 (REGENIE default) and minor allele frequency <1% and included age, sex, and the first 10 PCs as covariates. Genetic variants with $P < 1 \times 10^{-8}$ were declared ExWAS significant[46]. In addition to using European participants, recent ExWAS studies advocate to include participants of all ancestries[47,48]. Hence, we performed both white only and pan-ancestry analyses (added an additional covariate for four major ancestral groups, i.e., European, South Asian, East Asian, and African, identified by the K-Means clustering algorithm based on the first 10 PCs of genetic ancestry).

For gene-based association tests, we used SAIGE-GENE[26], a generalized mixed model approach that can adjust for both population stratification and genetic relationship. It performs rare-variant collapsing/aggregation tests, such as SKAT-O[49], burden[50] and SKAT[51,52]. We used predicted loss of function (pLOF) variants as the variants for gene sets. We defined pLOF variants as: stop gained, stop lost, start lost, splice donor, splice acceptor and frameshift based on the VEP[53] annotation and gnomAD pLOF variants[54]. We included age, sex, and the first 10 PCs as covariates. Genes with $P < 2.5 \times 10^{-6}$ were declared significant. We performed both white-only and pan-ancestry analyses (further added dummy variables for the major ancestral groups to the covariates).

### Glaucoma lookup in UKB and FinnGen

Since lowering IOP is currently the only glaucoma treatment, we performed a lookup in glaucoma traits in UKB and FinnGen resources for all ExWAS significant IOP rare variants and genes. In the UKB participants, glaucoma cases were identified if they self-reported glaucoma (UKB data fields 6148, 20002) or had an ICD-10 or ICD-9 diagnosis code for glaucoma (UKB data fields 131186, 131188, 41202, 41204, 41076, 41078, 41270), excluding glaucoma secondary to eye trauma, secondary to eye inflammation, secondary to other eye disorders, secondary to drugs, and other glaucoma. The selection of glaucoma based on self-reports and ICD-10 codes has been shown to be effective in previous studies[4,33]. Furthermore, the proportion of non-POAG cases in UKB was expected to be small[55]. Controls were identified as those who did not have glaucoma or self-reported eye problems. Overlapping with WES data, 14,378 white cases and 409,571 white controls, 15,606 pan-ancestry cases and 437,417 pan-ancestry controls remained. We further checked our top IOP genes in three FinnGen GWAS summary statistics, i.e., glaucoma, POAG, and use of antiglaucoma preparations and miotics (Freeze 7)[39], by querying each of them in their online results (see Data availability).

### Phenome-wide associations

For checking broader phenome-wide associations, we used PhenoScanner[56,57] and PheWeb[58]. PhenoScanner consists of over 5000

genetic association datasets from NHGRI-EBI, NHLBI and UKB results. We performed a query of all IOP associated genes to generate associations with glaucoma topical treatment phenotypes (online query default cutoff $P < 1.0 \times 10^{-5}$, GWAS results source: http://www.nealelab.is/uk-biobank). Supplementary Data 1 shows details of the queried eye-related phenotypes from PhenoScanner, among which there are 15 unique glaucoma topical treatments. It has been reported that dry eye and glaucoma often occur together[59]. Hence, significant artificial tear medication associations were also included. PheWeb uses summary statistics from the UKB to catalog millions of genetic markers across 1,403 binary traits. IOP associated genes queried in PheWeb generated a list of associations sorted by p-value. Out of these associations, phenotype traits related to the eye, cardiovascular, and nervous system were extracted. If no phenotype related to these traits were present, the association with the lowest p-value was reported.

### Gene expression

We used Genevestigator[16], a web-based gene expression database, to query bulk RNA information in different human tissues. Expression profiles of the queried genes in eye tissues from 210 human eyes were displayed in box plots showing the level of expression. For scRNAseq expression profiles, we used the Cell atlas of AHOP[18] (queried through Spectacle[60]) and of OAS[17] (queried through the Broad Institute Single Cell Portal [see Web Resources]) online databases. AHOP and OAS data were generated from seven and eleven human samples, respectively[17,18]. Each gene was queried to generate a heatmap and a violin plot displaying expression of various cell types related to AHOP and OAS of the eye, respectively.

### Drug targets prioritization

To prioritize drug targets for the identified rare-variant genes, we used the Open Targets online resource. For the identified genes, we queried the Open Targets for known drugs, their mechanisms of action (source ChEMBL), and disease information. The druggable genes provide key information on the relevance of these genes on IOP and glaucoma management and potential drugs for repurposing.

### rvPRS

From the pan-ancestry single-variant association results, we selected rare variants with $P < 5 \times 10^{-7}$ excluding intronic and synonymous variants. We assigned weights to these variants based on biological functions similar to that reported by Curtis[35,47]. Details of these variants and their weights are shown in the Supplementary Table 1. We then constructed a weighted rvPRS using PLINK similar to our previous approach[22], which was calculated as the summation of the number of rare risk alleles weighted by their biological functions. We then tested the association between the standardized rvPRS (subtracted the mean and divided by SD) and glaucoma in independent UKB white participants, who did not participate in the IOP measurements, using logistic regression adjusting for age and sex.

### Reporting summary

Further information on research design is available in the Nature Portfolio Reporting Summary linked to this article.

## Data availability

The UK Biobank data, both phenotypic and genetic, used in this study are available in the UK Biobank database and was accessed under application number 23424 (https://www.ukbiobank.ac.uk). The intraocular pressure summary statistics generated in this study are available at https://github.com/xraygao/GWAS_results. The following are links to public datasets we used in this study: ChEMBL, https://www.ebi.ac.uk/chembl/. FinnGen, https://www.finngen.fi/. Genevestigator, https://genevestigator.com/. PhenoScanner, http://www.phenoscanner.medschl.cam.ac.uk/. PheWeb, https://

pheweb.sph.umich.edu/. Spectacle, https://singlecell-eye.org/app/spectacle/. The Broad Institute's Single Cell Portal, https://singlecell.broadinstitute.org/single_cell/study/SCP1841/. UK Biobank, https://www.ukbiobank.ac.uk.

## Code availability

No original code was generated during this project. We followed online manuals on how to run each software. The programs used for data analysis: ANNOVAR, http://annovar.openbioinformatics.org/. REGENIE, https://rgcgithub.github.io/regenie/. SAIGE, https://github.com/weizhouUMICH/SAIGE. VEP, https://useast.ensembl.org/info/docs/tools/vep/index.html. R, https://www.r-project.org. PLINK, https://www.cog-genomics.org/plink/.

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

## Acknowledgements
We would like to thank the study participants and investigators from the UK Biobank as well as the staff who aided in data collection and processing. We also want to acknowledge the participants and investigators of the FinnGen study.

## Author contributions
X.R.G. conceived, planned and oversaw the present study. X.R.G. and M.C. analyzed the data. M.C. and A.J.A. performed the web queries. X.R.G., M.C., and A.J.A. wrote the manuscript.

## Competing interests
The authors declare no competing interests.
