## [Peer Review File · Nature Communications]

Whole-exome sequencing study identifies rare variants and genes associated with intraocular pressure and glaucomaREVIEWER COMMENTS

Reviewer #1 (Remarks to the Author):

The authors perform a whole exome wide study of IOP in the UKBB. They assess the relation between the top variants and glaucoma in UKB and between the top variants of glaucoma related traits in the FinnGen study. They use Open Targets to identify drugs that might be related to these loci and derive a rvPRS score for IOP.

Abstract:

It is misleading to suggest that whole exome data on IOP comes from 454,756 participants since IOP was only measured on 120,000 participants. Ultimately 98,674 white subjects and 110,260 pan-ancestry (all ancestry combined) subjects contributed to the WES.

Introduction:

Rather than say many IOP are associated with glaucoma it would be better to reference published bivariate global genetic correlations between IOP and glaucoma. These calculations indicated the shared heritability between IOP and POAG are >50%.

The statement that study rare variants can lead to drug targets for disease has no context. The example related to PCSK9 as a target for LDL cholesterol. Are there any examples more relevant to ophthalmology and glaucoma?

Results:

Based on Table 1, it looks like there are only 13, not 14 rare variants. Only 10 are novel, not 11. The Manhattan plots are of poor quality. They try to encompass the MYOC variant which is far more significant than the other variants which are barely WES-wide significant. As a result the annotation of the WES-wide hits are poor. It would be better to reduce the scale of the y-axis and just make a break point notation for MYOC being above the scale.

The PI should clearly state that the relation between ADRBI and IOP was found on gene-based testing only. Furthermore, beta-blocker use is a major determinant of IOP in the UKB. The authors should adjust for beta blocker use and see if the gene-based test still show that the rare beta receptor gene is still associated with IOP. Data on beta-blocker use is available in UKB.

The rvPRS does not include the MYOC variant because of its known large effect on glaucoma according to the authors. This is precisely why it should be included and the rvPRS recalculated. The effect size of the existing rvPRS in relation to glaucoma in UKB is not provided and the AUC for the rvPRS in predicting glaucoma is not performed. These analyses should be done and include the MYOC variant. Also, how much of the variance in IOP is explained by the rare variants. What is the relation between increasing IOP rvPRS and measured IOP in the UKB? How does these results compare to the IOP common variant PRS?

Methods:

Please provide the application number that allowed for access to the UKB resource.

The total number of glaucoma traits in FinnGen should be provided. It seems like there may be 100s or more of glaucoma traits because treatment with a myriad of glaucoma and non-glaucoma meds (like tear drops) are glaucoma traits and some of the weaker associations may not survive multiple comparisons. For example, treatment with timolol 0.25% as well as treatment with timolol 0.5% are considered separate traits.

Reviewer #2 (Remarks to the Author):

This study used WES data from 454,756 participants in UKB, to conduct an exome-wide association study (ExWAS) to identify rare variants and genes associated with IOP. Several novel SNPs were identified with putative IOP association and several novel genes were identified in gene-based analyses. However there are a number of concerns with this study including concerns about the methodology.

Not all 454,756 have IOP measurements and this sentence in the introduction is misleading: "Using WES data from 454,756 participants in UKB, we conducted an exome-wide association study (ExWAS) to identify rare variants and genes associated with IOP, evaluate their effects on glaucoma in UKB and the FinnGen cohort, and explore potential drug targets of the identified genes."

Minor allele frequencies are listed within the text and tables for variants with potential association however its not clear which population the allele frequency is referring to. For example rs201956837, noted to be associated with IOP in the pan-ancestry analysis is located near HLAB and has a minor allele frequency of 0.68% in Korean populations per dbSNP. The MAF listed in the text is 0.008% which is presumably the entire UB Biobank?

There is concern regarding the approach used for the pan-ancestry analysis. While the authors have used statistical methods for adjusting for population substructure this does not appear to have been adequate based on the findings. For example of the variants listed in Table 1 as 'additional hits' from the pan-ancestry analysis all but one have minor allele frequencies of at least 0.1% in populations other than European Caucasians and in particular 3 of these (CDK11A, PLAU, DPF3) have minor allele frequencies of 0.1% or greater in African populations and are not present in European ancestry, the main component of the UK Biobank. Because African ancestry is a strong risk factor for disease, there is potential for confounding with these ancestry specific variants and the analysis should be performed using only the African ancestry cases and controls from the UK biobank population rather than the entire overall cohort. Additionally, several of the ancestry-specific SNPs have much higher allele frequencies in Asian populations than what is reported in Table 1.

Additionally, the rvPRS which is comprised of SNPs from the pan-ancestry analysis may be spuriously identifying African ancestry rather than glaucoma risk in the analysis performed as 7 or more of the SNPs included in this score have highest allele frequencies in African ancestry populations.

Its also not clear why the authors removed MYOC from the rvPRS but not ANGPTL7 which is also relatively common and has large effects. This analysis should be repeated without ANGPTL7. Other than known genes MYOC and ANGPTL7, none of the SNPs or the corresponding genes (its not clear if the Pg is for SNP or for gene-based test) genes corresponding to the identified SNPs showed statistical evidence for association using either the criterion for single variant or gene-based tests. The one exception is HLAB but as discussed above this result could be a false positive based on population stratification.

The Top Pg designation in Table 1 for FINNGEN results is misleading as this could imply that this result is the top gene-based (or single variant) p value for the gene but is in fact the top gene-based p value for any glaucoma-related trait.

MTOR and ANGPTL7 are located in a genomic region with high LD suggesting that the reported gene-based MTOR result may not be due to causal variants in MTOR but in ANGPTL7. Can the authors provide additional data that confirms independent association for MTOR?

HLAB findings could be due to population substructure especially since this is only significant in pan-ancestry analyses and the HLA-B SNP associated with disease is found in 0.6% of Koreans.

Of the 19 significant single variants only 7 affect the protein coding sequence and 5 are intronic. Did the authors consider the role of these variants in alternative splicing?

Figure 1 is not possible to read and requires reformatting

Given the relatively small number of genes identified from either the single variant test or the gene-based tests it would be reasonable to examine their expression in published single cell RNA sequencing data to gain insight into their ocular expression.

Reviewer #3 (Remarks to the Author):

This paper presents an exome wide association study (ExWAS) for IOP using the UK Biobank resources. The authors focus on the role of rare variants in this complex trait which is a major risk factor and endophenotype for primary open angle glaucoma. They report novel genes and specific rare variants reaching exome-wide significance, show that a proportion of the genes and variants are also associated with glaucoma and related phenotypes in the UKB and FinnGen cohorts, and show that many of these genes have available drugs that may be relevant in the treatment of elevated IOP to prevent glaucoma onset or progression. There is a striking overlap with drugs used in the context of cardiovascular disease and hypertension.

Methods

Suggest changing the word "subjects" to "participants" or similar.

Variants with $MAC > 5$ are included in these analyses, but is there an upper limit on the MAF for inclusion? None of the significant variants have $MAF > 1\%$, but I'm not clear if that is because the dataset was limited to rare variants deliberately, or if the more common variants are just not significantly associated for some reason.

The Genevestigator tool is not noted in the methods. Please indicate what data source this is using and what was queried. While the PheWeb and PhenoScanner tools are named in the methods, some detail on how they were used and what the data source is would be appreciated, even if it is included in the Supplementary file with the appropriate table/figure.

Results

Demographics of the study cohort (other than self-reported ethnicity) are not presented (even though the analyses are adjusted for age and sex). As currently presented, the paper does not meet the requirements for the 'Sex and Gender Equity in Research – SAGER – guidelines' as required by the journal.

Supplementary Table S3 is noted for the first time in the Discussion. It should be described in the results first.

Only exome-wide significant results are presented. Are the full summary statistics to be made available?

Discussion

The SAIGE-GENE analysis uses only LoF variants. What limitations does this impose on the interpretation of the results? What advantages does this restriction bring to the analysis?

I noticed that the pan-ancestry analysis mostly detects non-coding variants while the white only analysis are mostly coding. Have the authors considered any sources of bias or confounding in the data that may lead to this?

The authors report a strong enrichment for drugs targeting cardiovascular related traits. Please discuss the relationship between IOP/glaucoma and hypertension/CVD in this context. Given the correlations between IOP and systemic blood pressure, has this analysis effectively detected variants contributing to blood pressure, and is that important in this context?

What are the limitations of and the rationale for assuming equal effect size for all variants in the rvPRS? The Betas in Supp Table S2 do not reflect equal effect sizes, and one variant has a negative effect.

Figures and Tables

Figure 1: Consider presenting the Manhattan plots excluding the MYOC variant, so that the resolution on the Y-axis can be clearer and it might be possible to read the labels added to the peak variants.

Table 1: Please add the protein variant annotations to the table for the coding variants.

Both Table 1 and Supp Table 2 say they contain single variants from the pan-ancestry analysis reaching exome-wide significance, however, there are considerably more variants in Supp Table 2 than in Table 1. Why is this? What is the difference between the tables?

In the Supplementary File, please put the legends with their appropriate figure, assuming this file will not be typeset for publication.

Supplementary Table S1: Please indicate in the descriptor for this table that it refers to association in the FinnGen data.

Re: Manuscript ID NCOMMS-22-22044-T title: “Whole-exome sequencing study identifies novel rare variants and genes associated with intraocular pressure and glaucoma”

Reviewer #1:

1. Abstract:

It is misleading to suggest that whole exome data on IOP comes from 454,756 participants since IOP was only measured on 120,000 participants. Ultimately 98,674 white subjects and 110,260 pan-ancestry (all ancestry combined) subjects contributed to the WES.

ANS: We apologize for our imperfect writing. We revised the abstract and changed “454,756” to “110,260”. We further added the sample size, $n = 312,825$, to the sentence “Furthermore, we constructed a rare-variant polygenic risk score and showed its significant association with glaucoma in independent participants ($n = 312,825$).” in the abstract.

2. Introduction:

Rather than say many IOP are associated with glaucoma it would be better to reference published bivariate global genetic correlations between IOP and glaucoma. These calculations indicated the shared heritability between IOP and POAG are >50%.

ANS: We thank the reviewer for the suggestion. We revised the sentence to “The bivariate genetic correlation between IOP and glaucoma was also found to be high (0.49)¹.”

3. The statement that study rare variants can lead to drug targets for disease has no context. The example related to PCSK9 as a target for LDL cholesterol. Are there any examples more relevant to ophthalmology and glaucoma?

ANS: We modified the sentence to “..., e.g., PCSK9 as a target for lowering low-density lipoprotein levels^{2,3} and MYOC as a target for gene therapy in treating myocilin-associated glaucoma⁴.”

4. Results:

Based on Table 1, it looks like there are only 13, not 14 rare variants. Only 10 are novel, not 11. The Manhattan plots are of poor quality. They try to encompass the MYOC variant which is far more significant than the other variants which are barely WES-wide significant. As a result the annotation of the WES-wide hits are poor. It would be better to reduce the scale of the y-axis and just make a break point notation for MYOC being above the scale.

ANS: We apologize for our counting error. We corrected the error and used the right counts in our revised manuscript. We thank the reviewer for the suggestion to use a break point notation in the Manhattan plots. We revised the figures and used a broken y-axis. The figures are much clearer now.

5. The PI should clearly state that the relation between ADRB1 and IOP was found on gene-based testing only. Furthermore, beta-blocker use is a major determinant of IOP in the UKB. The authors should adjust for beta blocker use and see if the gene-based test still show that the rare beta receptor gene is still associated with IOP. Data on beta-blocker use is available in UKB.

ANS: We thank the reviewer for the comment. We added the gene-based information to the description of ADRB1 on page 6. The sentence now reads “... ADRB1 (adrenoceptor beta 1, identified from our gene-based analysis) ...” We carried out a test to adjust for beta blocker use, the ADRB1 gene remained to have a significant association with IOP ($P = 5.73e-7$).

6. The rvPRS does not include the MYOC variant because of its known large effect on glaucoma according to the authors. This is precisely why it should be included and the rvPRS recalculated. The effect size of the existing rvPRS in relation to glaucoma in UKB is not provided and the AUC for the rvPRS in predicting glaucoma is not performed. These analyses should be done and include the MYOC variant. Also, how much of

the variance in IOP is explained by the rare variants. What is the relation between increasing IOP rvPRS and measured IOP in the UKB? How does these results compare to the IOP common variant PRS?

ANS: We thank the reviewer for the comment. In the revised manuscript, we included MYOC in our rvPRS construction. We further added effect size (odds ratio per standard deviation [SD]), AUC, and heritability estimation to the rvPRS paragraph on page 7: "This rvPRS is significantly associated with glaucoma with OR per SD = 1.12 and $P = 5.13 \times 10^{-16}$, indicating the relevance of these IOP rare variants in glaucoma." "Since the IOP heritability explained by WES rare variants is less than 2% (estimated using GCTA^{5,6}), the overall prediction improvement over the baseline model (with only age and sex) in terms of the area under the receiver operating characteristic curve (AUC) of the current rvPRS is relatively low, 0.5%, in comparison to more than 5% in AUC improvement from common variants⁷, which can explain about 40% of the IOP heritability⁸."

Regarding the relationship between IOP rvPRS and measured IOP in the UKB and its comparison to common-variant PRS (cvPRS), we ran a multiple linear regression, $IOP_{cc} \sim \text{age} + \text{sex} + \text{rvPRS (standardized)} + \text{cvPRS (standardized)}$, using 93,650 unrelated white participants. There is a positive relationship ($\beta = 0.14$, $P = 2.05e-41$) between rvPRS and IOP. Similarly, there is a positive relationship ($\beta = 1.08$, $P = 1.21e-2433$) between cvPRS and IOP. However, the difference between rvPRS and cvPRS need to be interpreted with care since the PRSs were calculated differently. The rvPRS and cvPRS were calculated using 25 rare WES variants and 1,697 common variants, respectively. Numerous reports, including ours, have been published on cvPRS. The best way to construct rvPRS, however, is not currently clear. Furthermore, we used rare variants from WES instead of whole-genome sequencing data, which is not fully available in UKB yet. In the Discussion, we added "The best way to construct rvPRS requires further investigation." in the study limitation.

7. Methods:

Please provide the application number that allowed for access to the UKB resource.

ANS: We added "application number 23424" to the UKB Resource paragraph in Methods on page 11.

8. The total number of glaucoma traits in FinnGen should be provided. It seems like there may be 100s or more of glaucoma traits because treatment with a myriad of glaucoma and non-glaucoma meds (like tear drops) are glaucoma traits and some of the weaker associations may not survive multiple comparisons. For example, treatment with timolol 0.25% as well as treatment with timolol 0.5% are considered separate traits.

ANS: We thank the reviewer for the comment. In the revised manuscript, we limited our query of FinnGen results to three glaucoma phenotypes and revised accordingly in Methods on page 14 "We further checked our top IOP genes in three FinnGen GWAS summary statistics, i.e. glaucoma, POAG, and use of antiglaucoma preparations and miotics (Freeze 7)⁹, by querying each of them in their online results (see data availability)." The glaucoma topical treatment results were queried through PhenoScanner. We added the detailed information regarding PhenoScanner in Methods on page 15. "PhenoScanner consists of over 5000 genetic association datasets from NHGRI-EBI, NHLBI and UKB results. We performed a query of all IOP associated genes to generate associations with glaucoma topical treatment phenotypes (online query default cutoff $P < 1.0 \times 10^{-5}$, GWAS results source: <http://www.nealelab.is/uk-biobank>). Supplementary Table S1 shows details of the queried eye-related phenotypes from PhenoScanner, among which there are 15 unique glaucoma topical treatments. It has been reported that dry eye and glaucoma often occur together¹⁰. Hence, significant artificial tear medication associations were also included." The multiple testing is not as bad as it appears (when all information is mixed together and unsorted) since there are only 15 unique glaucoma topical treatment phenotypes in our query.

Reviewer #2:

1. Not all 454,756 have IOP measurements and this sentence in the introduction is misleading: "Using WES data from 454,756 participants in UKB, we conducted an exome-wide association study (ExWAS) to identify rare variants and genes associated with IOP, evaluate their effects on glaucoma in UKB and the FinnGen cohort, and explore potential drug targets of the identified genes."

ANS: We apologize for our imperfect writing. We changed "454,756" to "110,260" in the sentence.

2. *Minor allele frequencies are listed within the text and tables for variants with potential association however its not clear which population the allele frequency is referring to. For example rs201956837, noted to be associated with IOP in the pan-ancestry analysis is located near HLAB and has a minor allele frequency of 0.68% in Korean populations per dbSNP. The MAF listed in the text is 0.008% which is presumably the entire UB Biobank?*

ANS: We apologize for our inaccurate description of minor allele frequency for this variant in our pan-ancestry results. It should be called A1Freq, allele 1 frequency in the analyzed sample. In pan-ancestry results, A1Freq is not equal to the MAF of any sub-population but a simple allele frequency of the analyzed sample. In the revised manuscript, we changed “MAF” to “AF” to avoid misunderstanding and added in the legend of Table1 for explaining A1Freq.

3. *There is concern regarding the approach used for the pan-ancestry analysis. While the authors have used statistical methods for adjusting for population substructure this does not appear to have been adequate based on the findings. For example of the variants listed in Table 1 as ‘additional hits’ from the pan-ancestry analysis all but one have minor allele frequencies of at least 0.1% in populations other than European Caucasians and in particular 3 of these (CDK11A, PLAU, DPF3) have minor allele frequencies of 0.1% or greater in African populations and are not present in European ancestry, the main component of the UK Biobank. Because African ancestry is a strong risk factor for disease, there is potential for confounding with these ancestry specific variants and the analysis should be performed using only the African ancestry cases and controls from the UK biobank population rather than the entire overall cohort. Additionally, several of the ancestry-specific SNPs have much higher allele frequencies in Asian populations than what is reported in Table 1.*

ANS: We thank the reviewer’s expertise in population allele frequency. As we mentioned in the discussion of our previous version of manuscript “the best approach for analyzing datasets of ancestrally diverse populations remains an ongoing research topic, especially for rare variants” and “One way to analyze multi-ethnic GWAS datasets is using meta-analysis, which is typically used for dealing with common variants. However, rare variants may not have enough carriers in individual ancestral groups, resulting in too few carriers to be analyzed.” For example, when we analyzed Black participants of the UK Biobank as suggested ($n = 3,286$ after overlapping with WES and IOP data), only 438,987 rare variants were in the output, leaving a lot of rare variants that could not be analyzed, while in contrast there were 2,624,332 rare variants in the output when we used white participants. Rather than leaving many potential important candidate rare variants not analyzable due to rare allele counts in any single sub-population or left out due to lack of statistical power, our pooled analysis utilized a mixed-effects model approach adjusting for genetic relationship matrix, principal components of genetic ancestry, and population clusters in a pan-ancestry framework, and provided a possible solution.

In the revised manuscript, we calculated allele frequencies and counts and analyzed all the significant rare variants from pan-ancestry single-variant analysis in sub-populations. Results are shown in Supplementary Tables S5 and S6 and added in the Discussion, “To further validate the robustness of our pan-ancestry results, we analyzed the significant rare variants in sub-populations, i.e. Black, Asian, and non-white all together, if there were sufficient rare-variant carriers for analysis, i.e. minor allele count (MAC) ≥ 5 (REGENIE default), in each sub-population. Supplementary Tables S5 and S6 show the sub-population-specific allele counts and association results, respectively. *CDK11A* rs556417493 is associated with IOP in Asian participants (AF = 0.08%, beta = 6.51, $P = 7.59 \times 10^{-9}$). *PLK5* rs776910868 is associated with IOP in Black participants (AF = 0.1%, beta = 8.14, $P = 8.42 \times 10^{-10}$). These sub-population-specific results are highly consistent with our pan-ancestry results. *ADSS2* rs375507039, *PLAU* rs367716060, and *DPF3* rs933632776 are associated with IOP in non-white participants and could not be analyzed in separate individual populations due to rare allele counts.” Again, these results are highly consistent with our pan-ancestry results, indicating the robustness of our pan-ancestry analysis.

4. *Additionally, the rvPRS which is comprised of SNPs from the pan-ancestry analysis may be spuriously identifying African ancestry rather than glaucoma risk in the analysis performed as 7 or more of the SNPs included in this score have highest allele frequencies in African ancestry populations.*

ANS: Though SNPs were extracted from pan-ancestry results, the actual association of rvPRS with glaucoma risk was done in white-only participants. Therefore, the concern of identifying African ancestry in

white-only participants is irrelevant.

5. *Its also not clear why the authors removed MYOC from the rvPRS but not ANGPTL7 which is also relatively common and has large effects. This analysis should be repeated without ANGPTL7. Other than known genes MYOC and ANGPTL7, none of the SNPs or the corresponding genes (its not clear if the Pg is for SNP or for gene-based test) genes corresponding to the identified SNPs showed statistical evidence for association using either the criterion for single variant or gene-based tests. The one exception is HLAB but as discussed above this result could be a false positive based on population stratification.*

ANS: To our knowledge, the association of *ANGPTL7* rs28991009 with IOP was first reported in our 2018 study (Supplementary Table 2)⁸. We previously selected variants with minor allele frequency (MAF) < 1% and $P < 5e-8$ for rvPRS construction. *ANGPTL7* rs28991009 had MAF = 0.796% and $P = 1.19E-10$. Hence, it was included in our rvPRS. Our previous exclusion of *MYOC* rs74315329 was not because the variant was relatively common but its highly significant association with IOP ($P = 1.22E-26$). We tried to avoid it masking other possible weaker associations and to demonstrate that the aggregated information of other rare variants was associated with glaucoma even when *MYOC* was not present. Excluding both *MYOC* and *ANGPTL7*, we still observed a significant association ($P = 1.85e-09$) between rvPRS and glaucoma in independent white participants, who did not participate in IOP measurements, using our revised rvPRS approach (details in the Methods section on page 16). Regarding whether to include *MYOC* and *ANGPTL7* in rvPRS, please see also Reviewer 1's comment 6.

The optimal p-value cutoff for selecting variants for constructing polygenic risk scores (PRS) can be different from the stringent genome-wide significance threshold, e.g., $5e-8$ for GWAS and $1e-8$ for ExWAS. For common variants, we found $5e-5$ was effective in our previous IOP PRS and POAG study⁷. In Khera et al.'s report, it required 6.6 million (most did not meet the stringent GWAS significance) and 5,218 SNPs ($P < 5e-4$) to reach optimal PRS performance for coronary artery disease and breast cancer, respectively. Currently, PRS using rare variants is not well studied. We found the p-value cutoff $5e-7$ was effective for selecting rare variants in the revised manuscript. We also stated in the limitation that "the best strategy to incorporate rare variants in PRS construction warrants further studies."

For the *HLA-B* variant rs201956837, there is consistent direction of allele effects and effect sizes in white (beta = 4.33, $P = 7.76 \times 10^{-6}$) and non-white (beta = 4.47, $P = 3.02 \times 10^{-4}$) (Supplementary Table S6 in the revised manuscript). Please note, *HLA-B* variant rs201956837 could not be analyzed in any single non-white sub-population due to limited rare-variant carriers. Using Fisher's method to combine the two p-values (white and non-white), we obtained $P = 2.35 \times 10^{-9}$, which is consistent with our pan-ancestry result, $P = 8.65 \times 10^{-9}$. These results indicate that the association is unlikely to be due to population stratification. Moreover, our rvPRS evaluation was carried out in white participants only, which was unlikely to be affected by population stratification. Furthermore, we removed intronic and synonymous variants from rvPRS in the revised manuscript and the intronic variant rs201956837 was no longer used in our rvPRS.

6. *The Top Pg designation in Table 1 for FINNGEN results is misleading as this could imply that this result is the top gene-based (or single variant) p value for the gene but is in fact the top gene-based p value for any glaucoma-related trait.*

MTOR and ANGPTL7 are located in a genomic region with high LD suggesting that the reported gene-based MTOR result may not be due to causal variants in MTOR but in ANGPTL7. Can the authors provide additional data that confirms independent association for MTOR?

ANS: We apologize for our imperfect notation. In the revised manuscript, we used clearer notation and limited the number of queried phenotypes in FinnGen to only three glaucoma-related traits, i.e. glaucoma, POAG, and antiglaucoma preparations and miotics. We also used distinct p-value notations to give clearer information on which phenotype the P refers to. For example, we used $P_G / P_P^+ / P_M^{++}$ to denote p-values for glaucoma, POAG, and antiglaucoma preparations and miotics, respectively.

Following the reviewer's suggestion, we conducted a gene-based analysis testing the association between *MTOR* and IOP conditional on *ANGPTL7* rs28991009 and found that the *MTOR* association remained significant with $P = 0.03$.

7. *HLAB findings could be due to population substructure especially since this is only significant in pan-ancestry analyses and the HLA-B SNP associated with disease is found in 0.6% of Koreans.*

ANS: For the *HLA-B* variant rs201956837, it met the stringent ExWAS significance, i.e. $1e-8$, in our pan-ancestry results. This does not exclude its association in white and non-white participants, though may not meet the stringent ExWAS significance threshold. For example, there is consistent direction of allele effects and effect sizes in white (beta = 4.33, $P = 7.76 \times 10^{-6}$) and non-white (beta = 4.47, $P = 3.02 \times 10^{-4}$) (Supplementary Table S6 in the revised manuscript). The association strength, $P = 7.76 \times 10^{-6}$ and $P = 3.02 \times 10^{-4}$, are also relatively strong given the reduced sample size in each sub-population. Please note, *HLA-B* variant rs201956837 could not be analyzed in any single non-white sub-population due to limited rare-variant carriers in the UK Biobank dataset (allele counts in Supplementary Table S5 in the revised manuscript). Using Fisher's method to combine the two p-values (white and non-white), we obtained $P = 2.35 \times 10^{-9}$, which is consistent with our pan-ancestry result, $P = 8.65 \times 10^{-9}$. These results indicate that the association is unlikely to be due to population stratification. We could choose to not analyze or report this variant. But by doing so we would be running the risk of ignoring important candidate rare variants for IOP. Koreans are not a major sub-population in the UK Biobank dataset and hence we could not evaluate the variant's association in Koreans. Therefore, in the Discussion section, we stated in the limitation of our study "Despite being the largest WES data currently available, diversity is still low, ... Other larger ancestral groups are no doubt invaluable and can provide further information for discovery and validation."

8. *Of the 19 significant single variants only 7 affect the protein coding sequence and 5 are intronic. Did the authors consider the role of these variants in alternative splicing?*

ANS: We thank the reviewer for the comment. We did not investigate alternative splicing since it is not our expertise. But we noticed that the rs201956837 in *HLA-B* is annotated as an intronic variant and an upstream transcript variant. Hence, we added in the Result section page 4 "Rs201956837 is an intronic variant as well as an upstream transcript variant." and added "upstream transcript variant" to the function of rs201956837 in Table 1 as well.

9. *Figure 1 is not possible to read and requires reformatting*

ANS: We thank the reviewer for pointing this out. We revised the figure and used a broken y-axis as suggested by Reviewer 1. The figures and gene labels are much clearer now.

10. *Given the relatively small number of genes identified from either the single variant test or the gene-based tests it would be reasonable to examine their expression in published single cell RNA sequencing data to gain insight into their ocular expression.*

ANS: We thank the reviewer for the suggestion. In the revised manuscript, we added a paragraph on gene expression in the Methods section on page 15 detailing the methods and databases we used. We also added in the Result section on page 5 "To seek biological support for the identified genes, we evaluated their gene expression using both bulk RNA and single-cell RNA (scRNA) expression datasets. Supplementary Figure S2 displays the bulk RNA expression information from Genevestigator¹¹. A number of genes, such as *BOD1L1*, *HLA-B*, *RPL10A*, and *RAB4B-EGLN2*, are highly expressed in the trabecular meshwork (TM). Several other genes, e.g., *ACAD10* and *DNTT*, show a medium gene expression in TM. Supplementary Figures S3 and S4 display the scRNA expression information from the Cell atlas of the human ocular anterior segment (OAS)¹² and of aqueous humor outflow pathways (AHOP)¹³, respectively. IOP and glaucoma related cell types can include TM fibroblasts, Schlemm canal endothelium (SCE), ciliary muscle (CM), corneal endothelium (CE), and vascular endothelium (VE)^{12,13}. Most of the identified genes show various levels of expression in these cell types. For example, *BOD1L1* is expressed in all the above cell types; *HLA-B* and *PTPRB* are expressed in SCE and VE; and *ACAD10* is expressed in CM and CE; and *RALYL* is expressed in CM, CE, and TM fibroblasts, to name a few."

Reviewer #3:

1. Methods

Suggest changing the word "subjects" to "participants" or similar.

ANS: We thank the reviewer for the suggestion. We changed “subjects” to “participants” in the Methods section.

2. *Variants with MAC>5 are included in these analyses, but is there an upper limit on the MAF for inclusion? None of the significant variants have MAF>1%, but I'm not clear if that is because the dataset was limited to rare variants deliberately, or if the more common variants are just not significantly associated for some reason.*

ANS: We only used rare variants with MAF < 1%. We modified the description to “We analyzed all rare variants with MAC ≥ 5 (REGENIE default) and minor allele frequency < 1% ...” in the Methods in the revised manuscript.

3. *The Genevestigator tool is not noted in the methods. Please indicate what data source this is using and what was queried. While the PheWeb and PhenoScanner tools are named in the methods, some detail on how they were used and what the data source is would be appreciated, even if it is included in the Supplementary file with the appropriate table/figure.*

ANS: In the revised manuscript, we added the description and citation for Genevestigator in the Gene Expression section in Methods on page 15. We also added a new paragraph for Phenome-wide associations explaining PhenoScanner and PheWeb on page 14 in the Methods section.

4. Results

Demographics of the study cohort (other than self-reported ethnicity) are not presented (even though the analyses are adjusted for age and sex). As currently presented, the paper does not meet the requirements for the ‘Sex and Gender Equity in Research – SAGER – guidelines’ as required by the journal.

ANS: We thank the reviewer for the comment. In the revised manuscript, we added the demographics of the study in the first paragraph of the Results section on page 4. “A total of 110,260 UKB participants were included in the IOP WES analysis, of which 98,674 were white. The mean (standard deviation [SD]) of age was 58 (8.1) years and 54% of the participants were female. The average IOP (SD) was 16.0 (3.4; range: 7.0 – 39.0) mmHg.” This improves the description of our research population and the suggestions outlined in SAGER guidelines¹⁴.

5. *Supplementary Table S3 is noted for the first time in the Discussion. It should be described in the results first.*

ANS: We added “Drug target genes *ADRB1*, *PTPRB*, and *RPL10A*, among others, were additionally found to have associations with vascular related phenotypes through PheWeb (Supplementary Table S3).” to the Results on page 7.

6. *Only exome-wide significant results are presented. Are the full summary statistics to be made available?*

ANS: The results will be available on the author's website, https://github.com/xraygao/GWAS_results, upon acceptance for publication.

7. Discussion

The SAIGE-GENE analysis uses only LoF variants. What limitations does this impose on the interpretation of the results? What advantages does this restriction bring to the analysis?

ANS: To our knowledge, there is no consensus on the best approach to analyze rare variants. We saw that many different approaches were used in the literature. Loss-of-function variants apparently have key biological functions. But they may not cover all possible signals. We think that this is a tradeoff between signal and noise and it can be phenotype dependent as well. Our single-variant analysis, which covered all types of rare variants, can be complementary as well. In the revised manuscript, we added in the limitation “There are many different approaches to analyze rare variants, e.g., grouping by predicted loss-of-function (pLOF) variants, missense variants, synonymous variants¹⁵, all inclusive¹⁶, and sliding window¹⁷. To our knowledge,

there is no consensus on the best approach to analyze rare variants. It can be phenotype dependent as well. Using other approaches may identify more rare variants and genes associated with IOP.”

8. *I noticed that the pan-ancestry analysis mostly detects non-coding variants while the white only analysis are mostly coding. Have the authors considered any sources of bias or confounding in the data that may lead to this?*

ANS: This is possibly due to chance. When we used a less stringent cutoff, i.e. 5×10^{-8} instead of 1×10^{-8} , the proportion of non-coding variants were similar, 8/24 vs. 3/10 for pan-ancestry vs. white. To further check the robustness of our pan-ancestry results, we conducted analysis in several sub-populations, i.e. black, Asian, and non-white. Please see our response to Reviewer 2’s comment 3. Our pan-ancestry and sub-population results are highly consistent, indicating the robustness of our pan-ancestry analysis.

9. *The authors report a strong enrichment for drugs targeting cardiovascular related traits. Please discuss the relationship between IOP/glaucoma and hypertension/CVD in this context. Given the correlations between IOP and systemic blood pressure, has this analysis effectively detected variants contributing to blood pressure, and is that important in this context?*

ANS: There is an established association between an increase in blood pressure affecting IOP¹⁸, and cardiovascular disease has been identified as an important risk factor for glaucoma¹⁹. Although there is a possibility, we are not attempting to conclude that our genes additionally contribute to blood pressure or cardiovascular disease. Instead, we are highlighting how our genes associated with IOP also have cardiovascular related drug targets, which continues to build the relationship between phenotypes pertaining to the eye and cardiovascular system. Therefore, the association offers potential in investigating these drugs targeting cardiovascular conditions for glaucoma. If these drugs show success in preventing cardiovascular conditions, then they may be repurposed for preventing glaucoma progression due to this genetic overlap.

For example, *ADRB1* was found to have an association with IOP. Drugs have been identified to target this gene to reduce blood pressure through its vascular modifications. There are also drugs targeting this gene that lower IOP by a yet to be confirmed mechanism. Since additional IOP associated genes have been identified to have this association with cardiovascular drug targets, these drugs may provide novel approaches to reducing IOP and preventing glaucoma progression.

10. *What are the limitations of and the rationale for assuming equal effect size for all variants in the rvPRS? The Betas in Supp Table S2 do not reflect equal effect sizes, and one variant has a negative effect.*

ANS: Assuming equal effect size is equivalent to unweighted RPS. Unweighted PRS and weighted PRS sometimes give similar results. But weighted PRS can be more powerful if the derived weights are close to the true values. In the previous version of the manuscript, the Betas in Supp Table S2 were the actual Betas from association tests, we did not list the beta=1 for the risk allele used in the unweighted rvPRS construction. In the revised manuscript, we gave the risk alleles and the weights used in the rvPRS construction in Supplementary Table S2 to avoid confusion.

11. *Figures and Tables*

Figure 1: Consider presenting the Manhattan plots excluding the MYOC variant, so that the resolution on the Y-axis can be clearer and it might be possible to read the labels added to the peak variants.

ANS: We thank the reviewer for the suggestion. We revised the figure and used a broken y-axis. The figures and gene labels are much clearer now.

12. *Table 1: Please add the protein variant annotations to the table for the coding variants.*

ANS: We added the information to Table 1.

13. *Both Table 1 and Supp Table 2 say they contain single variants from the pan-ancestry analysis reaching exome-wide significance, however, there are considerably more variants in Supp Table 2 than in Table 1. Why is this? What is the difference between the tables?*

ANS: Variants in Table1 met the stringent ExWAS significance threshold, 1×10^{-8} . Variants in Supplementary Table 2 were selected using a much less stringent threshold for constructing a PRS score. We chose a less stringent cutoff for selecting variants for PRS (Supplementary Table 2) because we and others found that including variants that do not reach stringent genome-wide significance can improve association strength and prediction accuracy^{7,20}. In the revised manuscript, we used 5×10^{-7} for selecting variants for PRS construction in Supplementary Table 2 and hope the statistical significance cutoff difference, 1×10^{-8} and 5×10^{-7} , is much clearer.

14. *In the Supplementary File, please put the legends with their appropriate figure, assuming this file will not be typeset for publication.*

ANS: We previously had the Figure Legend on a single page. In the revised manuscript, we added Figure legend at the bottom of each figure.

15. *Supplementary Table S1: Please indicate in the descriptor for this table that it refers to association in the FinnGen data.*

ANS: We thank the reviewer for the comment. We added PhenoScanner data source for treatment with eye drops “GWAS results source: <http://www.nealelab.is/uk-biobank>” in the revised Supplementary Table S1 (supplementary information, page 5).

References

- 1 Aschard, H. *et al.* Genetic correlations between intraocular pressure, blood pressure and primary open-angle glaucoma: a multi-cohort analysis. *Eur J Hum Genet* **25**, 1261-1267 (2017).
<https://doi.org/10.1038/ejhg.2017.136>
- 2 Dewey, F. E. *et al.* Distribution and clinical impact of functional variants in 50,726 whole-exome sequences from the DiscovEHR study. *Science* **354** (2016). <https://doi.org/10.1126/science.aaf6814>
- 3 Abul-Husn, N. S. *et al.* Genetic identification of familial hypercholesterolemia within a single US health care system. *Science* **354** (2016). [https://doi.org/ARTN aaf7000](https://doi.org/ARTN%20aaf7000) 10.1126/science.aaf7000
- 4 Jain, A. *et al.* CRISPR-Cas9-based treatment of myocilin-associated glaucoma. *Proc Natl Acad Sci U S A* **114**, 11199-11204 (2017). <https://doi.org/10.1073/pnas.1706193114>
- 5 Yang, J. *et al.* Common SNPs explain a large proportion of the heritability for human height. *Nat Genet* **42**, 565-569 (2010). <https://doi.org/ng.608> [pii] 10.1038/ng.608
- 6 Yang, J. *et al.* Genetic variance estimation with imputed variants finds negligible missing heritability for human height and body mass index. *Nat Genet* **47**, 1114-1120 (2015). <https://doi.org/10.1038/ng.3390>
- 7 Gao, X. R., Huang, H. & Kim, H. Polygenic Risk Score Is Associated With Intraocular Pressure and Improves Glaucoma Prediction in the UK Biobank Cohort. *Transl Vis Sci Technol* **8**, 10 (2019).
<https://doi.org/10.1167/tvst.8.2.10>
- 8 Gao, X. R., Huang, H., Nannini, D. R., Fan, F. & Kim, H. Genome-wide association analyses identify new loci influencing intraocular pressure. *Hum Mol Genet* **27**, 2205-2213 (2018).
<https://doi.org/10.1093/hmg/ddy111>
- 9 Kurki, M. I. *et al.* FinnGen: Unique genetic insights from combining isolated population and national health register data. *medRxiv*, 2022.2003.2003.22271360 (2022).
<https://doi.org/10.1101/2022.03.03.22271360>
- 10 Nijm, L. M., De Benito-Llopis, L., Rossi, G. C., Vajaranant, T. S. & Coroneo, M. T. Understanding the Dual Dilemma of Dry Eye and Glaucoma: An International Review. *Asia Pac J Ophthalmol (Phila)* **9**, 481-490 (2020). <https://doi.org/10.1097/APO.0000000000000327>
- 11 Hruz, T. *et al.* Genevestigator v3: a reference expression database for the meta-analysis of transcriptomes. *Adv Bioinformatics* **2008**, 420747 (2008). <https://doi.org/10.1155/2008/420747>
- 12 van Zyl, T. *et al.* Cell atlas of the human ocular anterior segment: Tissue-specific and shared cell types. *Proc Natl Acad Sci U S A* **119**, e2200914119 (2022). <https://doi.org/10.1073/pnas.2200914119>
- 13 van Zyl, T. *et al.* Cell atlas of aqueous humor outflow pathways in eyes of humans and four model species provides insight into glaucoma pathogenesis. *Proc Natl Acad Sci U S A* **117**, 10339-10349 (2020). <https://doi.org/10.1073/pnas.2001250117>
- 14 Heidari, S., Babor, T. F., De Castro, P., Tort, S. & Curno, M. Sex and Gender Equity in Research: rationale for the SAGER guidelines and recommended use. *Res Integr Peer Rev* **1**, 2 (2016).
<https://doi.org/10.1186/s41073-016-0007-6>
- 15 Karczewski, K. J. *et al.* Systematic single-variant and gene-based association testing of thousands of phenotypes in 426,370 UK Biobank exomes. *medRxiv*, 2021.2006.2019.21259117 (2022).
<https://doi.org/10.1101/2021.06.19.21259117>
- 16 Curtis, D. Analysis of 200 000 exome-sequenced UK Biobank subjects illustrates the contribution of rare genetic variants to hyperlipidaemia. *J Med Genet* **59**, 597-604 (2022).
<https://doi.org/10.1136/jmedgenet-2021-107752>
- 17 Li, X. *et al.* Dynamic incorporation of multiple in silico functional annotations empowers rare variant association analysis of large whole-genome sequencing studies at scale. *Nat Genet* **52**, 969-983 (2020). <https://doi.org/10.1038/s41588-020-0676-4>
- 18 Leske, M. C. & Podgor, M. J. Intraocular pressure, cardiovascular risk variables, and visual field defects. *Am J Epidemiol* **118**, 280-287 (1983). <https://doi.org/10.1093/oxfordjournals.aje.a113634>
- 19 Marshall, H. *et al.* Cardiovascular Disease Predicts Structural and Functional Progression in Early Glaucoma. *Ophthalmology* **128**, 58-69 (2021). <https://doi.org/10.1016/j.ophtha.2020.06.067>
- 20 Khera, A. V. *et al.* Genome-wide polygenic scores for common diseases identify individuals with risk equivalent to monogenic mutations. *Nat Genet* (2018). <https://doi.org/10.1038/s41588-018-0183-z>

REVIEWER COMMENTS

Reviewer #1 (Remarks to the Author):

My concerns have been addressed

Reviewer #2 (Remarks to the Author):

In this revised manuscript the authors have addressed a number of questions and concerns. There are several additional questions:

1) There are few details regarding the sub-population analyses, in particular how were the subjects in each sub-population identified? Self-report or genetically? Was the subpopulation analysis carried out in each group separately? It would be helpful to have a table that shows the N for each subpopulation and the results for each variant for each subpopulation, ie not just the results for the subpopulation where there was a significant result.

2) It is not clear that the pan-ancestry analysis did increase power as the authors claim in the discussion (lines 175-176). For example, the CDK11A variant has a beta of 6.55 and P of 8.67E-9 in the pan-ancestry analysis but also has very similar results in the subpopulation analysis (beta = 6.51, P= 7.6E-9). Since this variant is only present in Asians, the results appear to be very similar regardless of the inclusion of subjects from other ethnicities.

3) Considering that the UKBB is primarily European Caucasian the power in this study to identify risk variants in other racial groups is limited and the authors should include this as a limitation in their discussion.

4) What does a dash (-) mean in Table 1?

Reviewer #3 (Remarks to the Author):

All my questions have been answered

Re: Manuscript ID NCOMMS-22-22044-B title: “Whole-exome sequencing study identifies novel rare variants and genes associated with intraocular pressure and glaucoma”

Reviewer #1 (Remarks to the Author):

My concerns have been addressed

Reviewer #2 (Remarks to the Author):

In this revised manuscript the authors have addressed a number of questions and concerns. There are several additional questions:

1) There are few details regarding the sub-population analyses, in particular how were the subjects in each sub-population identified? Self-report or genetically? Was the subpopulation analysis carried out in each group separately? It would be helpful to have a table that shows the N for each subpopulation and the results for each variant for each subpopulation, ie not just the results for the subpopulation where there was a significant result.

ANS: We apologize for lack of details regarding the sub-population analyses. To our knowledge, there is only one data field, 21000, in the UK Biobank dataset that provides the sub-population information. In this manuscript, we focused on white-only and pan-ancestry analyses. A thorough analysis in each sub-population is beyond the scope of this single manuscript and can easily cause more questions similar to the reviewer asked. We did not infer each sub-population genetically since it is likely a standalone research project by itself. To identify sub-populations, we used the existing self-reported UK Biobank data field 21000, ethnic background, which matches closely to the reviewer’s description of sub-populations. The sub-population analysis was done in Black and Asian separately. We had already provided the observed counts (OBS_CT) in Supplementary Table S5 for the sub-populations analyzed. The $N = \text{observed (allele) count}/2$. We had provided the full results for the sub-populations Asian and Black in Supplementary Table S6 for the markers listed in Supplementary Table S5. Only two markers, rs556417493 in Asian and rs776910868 in Black, were analyzable and the rest of the markers was not analyzable due to low minor allele counts.

2) It is not clear that the pan-ancestry analysis did increase power as the authors claim in the discussion (lines 175-176). For example, the CDK11A variant has a beta of 6.55 and P of 8.67E-9 in the pan-ancestry analysis but also has very similar results in the subpopulation analysis (beta = 6.51, P= 7.6E-9). Since this variant is only present in Asians, the results appear to be very similar regardless of the inclusion of subjects from other ethnicities.

ANS: Statistical power is positively correlated with the sample size. A larger sample size gives greater power. Pan-ancestry has a larger sample size and can analyze more rare variants than any single sub-population. An online example of power and sample size can be found at the web link, https://www.mv.helsinki.fi/home/mjxpirin/GWAS_course/material/GWAS3.html. It is not about the special case of a single variant that is only polymorphic in a single sub-population, in which adding extra non-polymorphic individuals does not matter. We would like to emphasize the lines after 175-176, where we outlined reasons for our claim “It was evident that pan-ancestry analyses identified additional rare variants and genes beyond white-only analyses in both single-variant and gene-based analyses... Furthermore, the IOP rvPRS constructed using the rare variants identified from pan-ancestry analysis showed an even stronger association signal with glaucoma in independent white subjects than using white-only rare variants.”

3) Considering that the UKBB is primarily European Caucasian the power in this study to identify risk variants in other racial groups is limited and the authors should include this as a limitation in their discussion.

ANS: We want to highlight a sentence in the Discussion section where we believe we described this limitation of the low diversity of UKBB: “Despite being the largest WES data currently available, diversity is still low, and European subjects comprise about 94% of the UKB cohort. Other larger ancestral groups are no doubt invaluable and can provide further information for discovery and validation.”

4) What does a dash (-) mean in Table 1?

ANS: We apologize for the confusion. We added in the legend of Table 1 “To facilitate visualization, p-values that do not pass a pre-defined cutoff, 0.1 for UKB Glaucoma and 1×10^{-5} for PhenoScanner, are displayed as a dash (-).”

Reviewer #3 (Remarks to the Author):

All my questions have been answered

REVIEWERS' COMMENTS

Reviewer #2 (Remarks to the Author):

The response to the review addresses many of my concerns, however the authors should include using self-report to assign ethnic/racial groups as a limitation as this is well-known to be inaccurate. Appropriate racial designation is important for rare variant analyses.

REVIEWERS' COMMENTS

Reviewer #2 (Remarks to the Author):

The response to the review addresses many of my concerns, however the authors should include using self-report to assign ethnic/racial groups as a limitation as this is well-known to be inaccurate. Appropriate racial designation is important for rare variant analyses.

ANS: We thank the reviewer for the comment. For white subjects and the four major ancestral groups in pan-ancestry analyses, we used genetically defined ancestral groups. The self-reported ethnic/racial groups only apply to Black and Asian in Supplementary Table S3 and Supplementary Data S3. In the Methods section on page 13, we added, “(also removed outliers with genetic ancestry at least six SDs from the means of the first two PCs)” (SD: standard deviation, PC: principal component) after “98,674 white participants” in the third to the last sentence in the IOP Measurements in UKB section. On page 14, we added “based on the first 10 PCs of genetic ancestry” to the last sentence of the first paragraph explaining the four major ancestral groups in our pan-ancestry analyses in the Single-variant and Gene-based ExWAS Analyses section. We also added in the limitations of our study in the last sentence on page 10 and the first sentence on page 11 that “Sub-population labels for our analyses in Black and Asian in Supplementary Table S3 and Supplementary Data S3 were extracted from self-reported ethnicity (UKB data field 21000).” We hope the ethnic/racial information is much clearer now and this satisfies the reviewer.